# Thromboembolic Events with Cyclin-Dependent Kinase 4/6 Inhibitors in the FDA Adverse Event Reporting System

**DOI:** 10.3390/cancers13081758

**Published:** 2021-04-07

**Authors:** Emanuel Raschi, Michele Fusaroli, Andrea Ardizzoni, Elisabetta Poluzzi, Fabrizio De Ponti

**Affiliations:** 1Pharmacology Unit, Department of Medical and Surgical Sciences, Alma Mater Studiorum—University of Bologna, 40126 Bologna, Italy; michele.fusaroli2@unibo.it (M.F.); elisabetta.poluzzi@unibo.it (E.P.); fabrizio.deponti@unibo.it (F.D.P.); 2Medical Oncology Unit, Department of Experimental, Diagnostic and Specialty Medicine, Policlinico S. Orsola-Malpighi, Alma Mater Studiorum—University of Bologna, 40126 Bologna, Italy; andrea.ardizzoni2@unibo.it

**Keywords:** cyclin-dependent kinase (CDK) 4/6 Inhibitors, thromboembolism, pharmacovigilance, FAERS, signal

## Abstract

**Simple Summary:**

This post-marketing research addressed the role of cyclin-dependent kinase 4/6 inhibitors, relatively new anticancer drugs approved for advanced breast cancer in cancer-associated thrombosis. We used the Food and Drug Administration pharmacovigilance database to retrospectively assess thromboembolic events in the real world. A potential class effect was found for venous thrombosis, whereas distinctive arterial events emerged for ribociclib. These signals call for both prospective research and early proactive monitoring by oncologists, also in patients without apparent risk factors. These findings strengthen the role of timely pharmacovigilance to detect and characterize post-marketing adverse events of special interest, thus supporting patient care.

**Abstract:**

We analyzed thromboembolic events, recognized (AESIs), with cyclin-dependent kinase (CDK)4/6 inhibitors, using the Food and Drug Administration adverse event reporting system. Methods: Thromboembolic events were characterized in terms of spectrum [venous and arterial thromboembolism (VTE; ATE)] and clinical features by combining the disproportionality approach [reporting odds ratio (ROR) with 95% confidence interval (CI)] with individual case assessment. Results: A total of 1722 thromboembolic events were retained. Increased VTE reporting emerged for CDK4/6 inhibitors in the exploratory analyses (*n* = 659; ROR = 1.51; 95% CI = 1.39–1.63), with consistent disproportionality in the consolidated analyses (e.g., deep vein thrombosis with abemaciclib: 17; 1.98; 1.22–3.19). Higher-than-expected ATE reporting was found for ribociclib, including myocardial infarction (41; 1.82; 1.33–2.48), with rapid onset (median latency 1 vs. 6 months for other CDK4/6 inhibitors). Causality was highly probable or probable in 83.2% of cases, with a negligible proportion of pre-existing drug- and patient-related risk factors except for cardiovascular comorbidities (26%). Conclusions: Although causal association cannot be firmly inferred, oncologists should proactively monitor the occurrence of VTE with CDK4/6 inhibitors. The unexpected distinctive increased ATE reporting with ribociclib deserves urgent clarification though large comparative population-based studies. We support pharmacovigilance for the post-marketing characterization of AESIs, thus promoting real-time safe prescribing in oncology.

## 1. Introduction

The advent of cyclin-dependent kinase (CDK)4/6 inhibitors such as palbociclib, abemaciclib, and ribociclib are changing the therapeutic landscape of advanced breast cancer [1]. Notably, similarities and differences exist in terms of pharmacodynamics (e.g., selectivity for CDKs) and pharmacokinetics (e.g., lipophilicity) [2,3]. Although these subtle differences may be used to optimize treatment selection, all CDK4/6 inhibitors appear to have similar efficacy in both first- and second-line therapy [4]. Conversely, clinically relevant differences exist in the pattern and frequency of toxicities, thus making safety a key determinant in the physician’s treatment choice [5,6].

Cancer-associated thrombosis is a major cause of morbidity and mortality in patients with cancer, the most common type being venous thromboembolism (VTE). Several risk factors for VTE also coexist, including anticancer drugs (e.g., tamoxifen, angiogenesis inhibitors, and chemotherapy), with multiple mechanisms and unsettled issues regarding the optimal diagnostic, preventive, and management strategies [7].

Data on thromboembolism with CDK4/6 inhibitors are still preliminary and derive mainly from pre-approval randomized controlled trials (RCTs) [8,9]. Of note, VTE is an adverse event of special interest (AESI) for abemaciclib, and it is recognized with a specific warning on the label.

A recent pooled analysis of MONARCH 2 and 3 trials found that any-grade VTE, including pulmonary embolism (PE) and deep vein thrombosis (DVT), occurred in 4.8% and 6.1% of abemaciclib-treated patients (vs. 0.9% and 0.6% in the control groups). They were most commonly managed using low-molecular-weight heparin, with drug discontinuation in 4 patients (3 subjects died), and most of the patients had pre-existing risk factors which were balanced between the arms [10]. The latest long-term pooled analysis of 3 RCTs on the safety of palbociclib did not identify imbalances among groups in terms of PE [11]. Conversely, in the updated overall survival analyses of MONALEESA-3 and MONALEESA-7 studies on ribociclib, clear imbalances emerged: 23 PE events (4.8%; 2.3% grade 3/4) vs. 2 (0.8%; none of grade 3/4), and 9 PE events (2.7%; 1.5% grade 3/4) vs. 3 (0.9%; 0.6% grade 3/4), respectively [12,13].

Regarding post-marketing observational evidence, preliminary data on small cohorts have recently started to accrue. Gervaso et al. [14] retrospectively analyzed 424 patients and showed that VTE occurred in 9% of subjects (mainly receiving palbociclib): DVT alone was found as the most common presentation (47.4%) followed by visceral vein thrombosis (VVT) (21.1%) and PE (18.4%). A single-center audit of 64 individuals treated with palbociclib over a five-month period in an Irish tertiary referral hospital recorded seven venous thromboembolic events (11%) [15], whereas no increased risk of PE was found when comparing the propensity score matching new users of palbociclib–fulvestrant to historical users of fulvestrant monotherapy [16]. Finally, a recent case report described the occurrence of an acute myocardial infarction due to plaque erosion 2 weeks after abemaciclib treatment [17].

A question arises as to whether or not VTE should be considered a class effect of CDK4/6 inhibitors. In this context, international pharmacovigilance databases such as the Food and Drug Administration Adverse Event Reporting System (FAERS) have been successfully exploited for the accurate and timely real-world safety assessment of recently marketed anti-cancer drugs, especially to detect rare but serious adverse events (AEs) such as VTE, which may not be fully appreciated in the pre-marketing setting [18,19,20,21]. When properly designed, the accuracy of pharmacovigilance analyses through FAERS is noteworthy (i.e., the ability to actually distinguish between true and false negatives) [22], and a recent study found that risk estimates from meta-analyses and pharmacovigilance analyses were correlated in some cases [23], thus supporting the role of FAERS in designing targeted pharmaco-epidemiological studies or exploring the underlying pharmacological bases [24].

On these grounds, we aimed to comprehensively characterize thromboembolic AEs with CDK4/6 inhibitors submitted to FAERS in terms of spectrum, clinical features, and potential causality (i.e., the probability of drug-related contributions).

## 2. Materials and Methods

### 2.1. Data Source and Study Design 

This observational, retrospective pharmacovigilance analysis combined disproportionality approaches and case-by-case evaluation of thromboembolic AEs recorded in the FAERS archive. To this purpose, publicly available quarterly data were downloaded (https://fis.fda.gov/extensions/FPD-QDE-FAERS/FPD-QDE-FAERS.html, accessed on 6 April 2021), pre-processed to remove duplicates (i.e., reports overlapping in key pre-specified fields, including active substance(s), AEs, event date, age, gender, reporter country, weight), and restricted to the period from 2015 (the first-in-class palbociclib was approved in February 2015) to September 2020 (the latest available data). Exposure assessment considered drugs recorded as suspect and concomitant. 

We first performed the so-called disproportionality approach. If the proportion of AEs was greater in patients exposed to a specific drug or drug class such as CDK4/6 inhibitors (cases) than in patients not exposed to this drug (non-cases), a disproportionality signal emerged [25]. Through this so-called case/non-case approach, the reporting odds ratio (ROR) with relevant 95% confidence interval (95% CI) was calculated and deemed statistically significant by common thresholds (i.e., the lower limit of the 95% CI > 1 with at least 3 cases) [26].

Cases (i.e., thromboembolic events) were identified using the following Standardized Medical Dictionary for Regulatory Activities Queries (SMQs): “embolic and thrombotic events” (comprehensive search), “embolic and thrombotic events, venous,” “embolic and thrombotic events, arterial,” and “embolic and thrombotic events, other.” These SMQs allowed for a high-sensitivity search while the individual signs and symptoms, named preferred perms (PTs), offered a clinical perspective by specifically describing the nature and origin of the event. For instance, DVT is included among the VTEs, myocardial infarction is classified as arterial thromboembolism (ATE), and the category “other” comprises thromboembolic events with unknown origin or nature (e.g., cerebrovascular accident, atrial thrombosis, embolism). The full list of PTs within relevant SMQs is provided in Appendix A.

### 2.2. Disproportionality Analyses

A stepwise approach was implemented *a priori* to minimize major confounders and biases, taking into account Good Signal Detection Practices in pharmacovigilance [27]:An *exploratory disproportionality approach* comparing CDK4/6 inhibitors with all other drugs reported in the FAERS database and using tamoxifen as a positive control (well-known association with thrombosis).A *consolidated disproportionality approach* comparing CDK4/6 inhibitors with other oncological drugs (using AEs recorded for at least one anticancer agent), a recommended strategy to provide a clinical perspective (i.e., selecting a real-world subpopulation that presumably shares a set of common risk factors) while reducing the so-called indication bias by considering the susceptibility of oncological patients to thrombosis [25]. Moreover, competing AEs potentially masking the identification of thromboembolic events were removed (i.e., diarrhea, agranulocytosis, torsade de pointes, and interstitial lung disease) using relevant SMQs (broad search) [28]. Analyses were performed at the SMQ and PT levels through the open-source R software (version 4.0.2; 22 June 2020).

### 2.3. Case-by-Case Assessment

Thromboembolic events were described in terms of the following demographic characteristics: age, reporter country (US, Europe, Asia), reporter type (e.g., clinician vs. consumer), fatality (i.e., death reported as the outcome), and seriousness (focusing on events resulting in hospitalization). The following clinical features were inspected: latency (i.e., time to onset expressed in days with interquartile range (IQR), calculated as the difference between the start of therapy and the date the event occurred), dechallenge (clinical improvement after the offending agent was suspended), rechallenge (occurrence of a similar reaction after re-administration, usually unintentional), presence of metastasis, neoplasm progression, co-reported hormone therapy, anemia and cardiovascular comorbidities (based on co-reported cardiovascular drugs and/or cardiovascular indications).

Individual cases were assessed for causality (categorized as highly probable, probable, possible, unlikely) according to an adaptation of the standardized WHO–UMC system, a probabilistic algorithm (https://www.who.int/medicines/areas/quality_safety/safety_efficacy/WHOcausality_assessment.pdf; last accessed date: 6 April 2021). Highly probable cases were those with plausible time to onset, alternate drugs ruled out, and positive dechallenge and/or rechallenge.

To this purpose, the following drugs were identified as a risk factor for thrombosis (by having strong evidence of thromboembolic risk) or being a proxy of a disease associated with thrombosis susceptibility: contraceptives/estrogens/progestogens, glucocorticoids, antidepressants, antidiabetics, angiogenesis inhibitors, erythropoiesis-stimulating agents. Moreover, concomitant antithrombotic drugs (antiplatelet agents, heparins, vitamin K antagonists, direct oral anticoagulants) were checked as potential proxies of pre-existing thromboembolic risks/events, as or indicative of management strategies (if the date of administration followed the onset date of the thrombotic event).

## 3. Results

### 3.1. Descriptive Analyses

Overall, 5,911,056 reports were retained after FAERS processing, of which 49,125 with CDK4/6 inhibitors (0.83%). A total of 228,443 thromboembolic events were found, of which 1722 recorded CDK4/6 inhibitor use (0.75% of total FAERS thromboembolic AEs, and 3.5% within AEs to CDK4/6 inhibitors), a number which increased steadily over time (Table 1). Of note, CDK4/6 inhibitors were almost exclusively reported as suspect, with a large contribution from consumers, a common feature in FAERS. Hospitalization and death were recorded in 52.8% (vs. 16.9% of non-thromboembolic events) and 12.5% (vs. 10.8%) of cases, respectively. 

The median number of events per thromboembolism report was 3 (64.8% of reports with <4 co-reported events), mainly fatigue (*n* = 239), decreased white blood cell count (147), nausea (134), tumor progression (130), dyspnea (129) and diarrhea (120). The median number of medications per thromboembolism report was 2 (68.5% of reports with <4 co-reported drugs). A higher proportion of cardiovascular comorbidities, mainly hypertension, was found among the cases, than in the non-cases (26% vs. 15.7%, respectively), ranging from 16% with abemaciclib in ATE, to 30% with palbociclib in VTE.

### 3.2. Disproportionality Analyses

Increased reporting of thromboembolic AEs (comprehensive search) emerged only for ribociclib in the exploratory analysis (*n* = 254; ROR = 1.18, 95% CI = 1.04–1.34). Consistent disproportionality signal was found for VTE and CDK4/6 inhibitors as a class and as individual drugs in both the exploratory (659; 1.51; 1.39–1.63) and consolidated analyses (659; 1.09; 1.01–1.18), except for palbociclib in the exploratory analysis (512; 0.99; 0.91–1.09) (Table 2). No significant disproportionality emerged for ATE, but ribociclib generated a disproportionality signal for “thromboembolic events (other)” (104; 1.25; 1.02–1.51).

Consistent disproportionality emerged for all the CDK4/6 inhibitors on PE, with significantly increased reporting of DVT with abemaciclib (17; 1.98, 1.22–3.19), cerebrovascular accidents with palbociclib (248; 1.22; 1.07–1.38), and myocardial infarction with ribociclib (41; 1.82; 1.33–2.48) (Figure 1). Significant RORs were also found for the composite DVT and/or PE for all CDK4/6 inhibitors: abemaciclib (41, 1.97, 1.45–2.69), palbociclib (369, 1.15, 1.03–1.27), and ribociclib (73, 1.85, 1.46–2.33). No disproportionality emerged for VVT: only eight cases of portal vein thrombosis were recorded with palbociclib, and six of retinal and jugular vein thrombosis, without reaching statistical significance. The full list of reported thromboembolic AEs, with relevant disproportionality, is provided in Appendix A.

### 3.3. Case-by-Case Assessment

Causality assessment was highly probable or probable in 83.2% of cases, with similar percentages among CDK4/6 inhibitors (Table 3). Only in a negligible proportion of cases were drug- and disease-related risk factors recorded, namely glucocorticoids and antidepressants (7% and 8% of palbociclib VTE cases, respectively). Antithrombotic drugs and denosumab were found in 16% and 9% of thromboembolic events, respectively. In 27.3% of the cases (75/275), available data regarding time and indications did allow us to assert that antithrombotic agents were started after the thromboembolic event. A lower time to onset emerged for ribociclib in ATE (27 days; IQR = 16–134) than with other CDK4/6 inhibitors (more than 200 days), with Germany as the main reporter country. Drug interruption was recorded in 38.6% of cases, with a positive dechallenge in 39.2% of abemaciclib VTE cases. The full description of thromboembolic AEs is provided in the Appendix A.

## 4. Discussion

Comparative safety and tolerability are a current priority with CDK4/6 inhibitors, and the various toxicities differ in terms of the spectrum and timing of occurrence: neutropenia mainly occurs with palbociclib during the first three cycles of treatment, diarrhea occurs with abemaciclib in 6–8 days, whereas QT prolongation and liver injury mainly occur with ribociclib. These safety issues impact drug choice and monitoring, also considering patient susceptibility and factors influencing pharmacokinetics/pharmacodynamics, including ethnicity [6,29,30]. Thromboembolism, mainly in the form of VTE, emerged as AESI in pivotal RCTs, especially for abemaciclib, which received a specific warning in the label.

This large-scale contemporary pharmacovigilance analysis further enlightened the complex safety profile of CDK4/6 inhibitors by characterizing, for the first time, their global reporting of thromboembolic events. Oncologists should be aware that thrombosis does occur with all CDK4/6 inhibitors, even in patients without apparent drug-related risk factors, thus posing a challenge to management strategies, including prophylactic approaches. These findings carry remarkable implications for the evolving uses of CDK4/6 inhibitors, including potential combinations with tamoxifen and extended use in the adjuvant setting, where even a small increased thrombotic risk may influence the delicate risk-benefit balance. 

Overall, a variegated spectrum of vascular manifestations emerged, and thromboembolic events represented 3.5% of AEs recorded with CDK4/6 inhibitors. Of note, as compared to other rare AEs such as interstitial lung disease, thromboembolism was not affected by regulatory warnings. We found increased reporting of VTE for all CDK4/6 inhibitors, especially with PE, which was consistent in both the exploratory and consolidated disproportionality analyses, except in the case of palbociclib.

Abemaciclib was associated with a stronger and robust disproportionality signal of DVT, which was in line with evidence from RCTs [10], including the recent monarchE study in node-positive, high-risk, early breast cancer, where venous thromboembolic events were reported in 2.3% of patients (63 events) in the abemaciclib arm and 0.5% in the control arm (1.1% vs. 0.2% for DVT) [31]. Palbociclib was the only associated with elevated reporting of cerebrovascular accidents, although the interpretation of these data is uncertain. While no imbalances emerged from the long-term pooled safety of 3 RCTs and a real-world cohort study [11,15], we found a non-negligible proportion of cardiovascular comorbidities in thromboembolic events with palbociclib in FAERS (30% in VTE). 

Notably, we found a potential concern in terms of ATE reporting, namely myocardial infarction with ribociclib. Significant disproportionalities emerged also for transient ischemic attacks, cerebral ischemia, paraplegia, and paraparesis (Appendix A), with a more rapid time to onset than other CDK4/6 inhibitors (1 vs. 6 months, respectively). The case-by-case evaluation failed to detect concomitant drug-or disease-related risk factors, except for cardiovascular comorbidities (26% in ATE).

Overall, the higher proportion of cardiovascular comorbidities found for palbociclib and ribociclib, as compared to abemaciclib, raised the hypothesis of a channeling use, i.e., preferential prescribing of ribociclib/palbociclib to susceptible patients due to perceived higher thrombotic risk with abemaciclib. The latest retrospective analysis of 266 patients found a 10.9% rate of thrombosis events, with highest incidence of ATE being with palbociclib and ribociclib. The fact that the Khorana score was not predictive of thrombosis further supports a pro-thrombotic class effect with CDK 4/6 inhibitors, although this score was not validated for oral anticancer drugs and did not consider metastasis status [32].

Taken together, these data call for constant epidemiological surveillance and urgent clarification by means of large, population-based studies and dedicated RCTs. The mechanisms and risk stratification of this pro-thrombotic liability are uncertain, and dedicated analyses of the mutation profiles are needed for a personalized approach. We support early vigilance for VTE and ATE, and cardiological consultation in patients with cardiovascular risk factors, including hypertension, coronary artery disease, previous episodes of thromboembolism, recent surgical interventions, immobilization, old age, heart failure, trauma with fractures, obesity, and hereditary and acquired thrombophilic disorders. Of note, a history of angina pectoris and recent myocardial infarction were among the exclusion criteria of pre-approval RCTs of ribociclib. Long-term monitoring for VTE is also needed, especially for the timely diagnosis of signs and typical symptoms such as edema, sudden leg swelling, thrombophlebitis, shortness of breath, hypoxia, chest pain, polypnea, and tachycardia.

We acknowledge the limitations of our study, including no certainty on causation, incomplete or missing information, potential remaining duplicates, and lack of data on population exposure, which did not allow us to provide incidence rates and risk rankings based on the ROR strength [26]. Moreover, clinical features (body mass index, patient histories, comorbidities) and instrumental assessments (ultrasonography and laboratory parameters) to support diagnosis, grading of severity, and prediction of the underlying thrombotic risk (e.g., the Khorana risk score) cannot be fully analyzed. Finally, the contribution of additional drugs with underlying thrombotic liability cannot be excluded, including aromatase inhibitors and/or fulvestrant. Therefore, a causality assessment approach remains probabilistic. Nonetheless, our real-world analysis contributed to the cumulative knowledge about the safety of CDK4/6 inhibitors in an unselected population using a global database and combining a disproportionality approach with case-by-case evaluation, thus supporting the generalizability of findings and the potential existence of an underlying drug-related component. There are no clues to support the existence of major distortions to the data such as confounding by indication and notoriety bias. We implemented *a priori* major strategies to increase the accuracy of the disproportionality analysis [22], thus supporting pharmacovigilance as a potential indicator of risk in the real world [23].

## 5. Conclusions

Increased VTE reporting raised the hypothesis of a potential class effect for CDK4/6 inhibitors, in contrast with the pre-approval evidence, and called for early and long-lasting proactive monitoring by onco-cardiologists.

The unexpected increased ATE reporting with ribociclib, to be interpreted with caution, deserves urgent and large comparative population-based studies to establish actual event rates, fully elucidate risk factors that might lead to proper risk management, and clarify the relationship with overall survival and potential predictors. 

We support the role of spontaneous reporting systems as a crucial source for the constant epidemiological surveillance and real-time characterization of AESIs, thus promoting the safe prescription of oncological drugs.

## Figures and Tables

**Figure 1 cancers-13-01758-f001:**
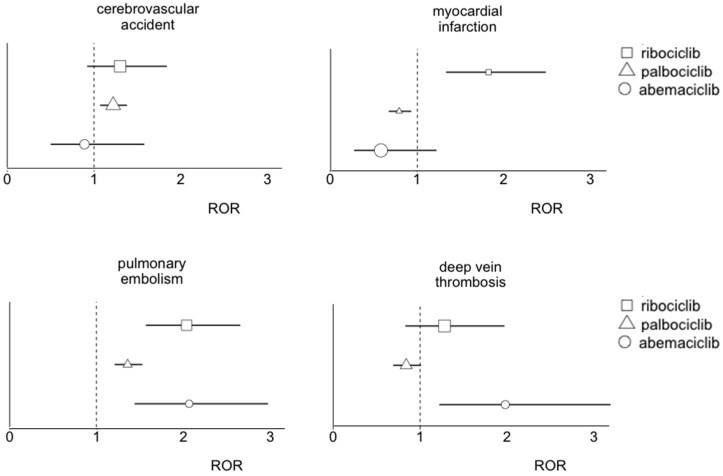
Consolidated disproportionality for selected thromboembolic events. The size of the ROR estimates is proportional to the number of cases.

**Table 1 cancers-13-01758-t001:** Demographic data of CDK4/6 inhibitors.

Features	Thromboembolic Events (Cases)	Other Events (Non-Cases)
Total reports	1722	47,403
*Age Distribution*		
Adult	554 (39.6%)	18,032 (47.7%)
18–29	4 (0.7%)	135 (0.7%)
30–49	104 (18.8%)	4437 (24.6%)
50–64	446 (80.5%)	13,460 (74.6%)
Elderly	844 (60.4%)	19,789 (52.3%)
65–75	501 (59.4%)	11,630 (58.8%)
76–85	276 (32.7%)	6549 (33.1%)
>85	67 (7.9%)	1610 (8.1%)
Other/Missing	324	9582
*Type of Reporter*		
Consumer	601 (36.2%)	17,489 (38.6%)
Physician	479 (28.9%)	8222 (18.1%)
Other	289 (17.4%)	8762 (19.3%)
Pharmacist	177 (10.7%)	7788 (17.2%)
Other healthcare professionals	114 (6.9%)	3037 (6.7%)
Lawyer	0 (0.0%)	6 (0.0%)
Missing	62	2099
*Year of Reporting*		
2015	56 (3.3%)	2021 (4.3%)
2016	170 (9.9%)	4868 (10.3%)
2017	316 (18.4%)	7877 (16.6%)
2018	348 (20.2%)	9593 (20.2%)
2019	473 (27.5%)	12,651 (26.7%)
2020 (up to Sept)	359 (20.2%)	10,393 (21.9%)
*Reporter Country*		
North America	1089 (63.2%)	38,172 (80.6%)
Europe	382 (22.2%)	4504 (9.5%)
Asia/Japan	132 (7.7%)	2509 (5.3%)
South America	100 (5.8%)	1829 (3.9%)
Oceania	12 (0.7%	225 (0.5%)
Africa	7 (0.4%)	144 (0.3%)
Missing	0	20
*Outcome* #		
Death	215 (12.5%)	5122 (10.8%)
Life-threatening	116 (6.7%)	518 (1.1%)
Disability	26 (1.5%)	194 (0.4%)
Required-Intervention	0 (0.0%)	61 (0.1%)
Congenital Anomaly	1 (0.1%)	12 (0.0%)
Hospitalization	910 (52.8%)	7991 (16.9%)
Other	1049 (60.9%)	15,391 (32.5%)
*Reported Causative Role*		
Primary Suspect	1401 (81.4%)	41,032 (86.6%)
Secondary Suspect	273 (15.9%)	5179 (10.9%)
Interacting	9 (0.5%)	79 (0.2%)
Concomitant	39 (2.3%)	1113 (2.3%)
*Clinical Features*		
Metastasis specified ±	653 (37.9%)	12,447 (26.3%)
Neoplasm progression	130 (7.5%)	3710 (7.8%)
Hormone Therapy ‡	913 (53.0%)	18,749 (39.6%)
Anaemia €	106 (6.2%)	2579 (5.4%)
Cardiovascular comorbidities $	448 (26.0%)	7427 (15.7%)
Weight, median (25–75%; missing)	72 (62–85; 610)	70 (60–82; 12,191)
*Time to Onset, Days (Interquartile Range; n with Available Data)*		
Abemaciclib	115 (44–265; 46) ɣ	52 (22–128; 1291)
Palbociclib	123 (44–330; 407) ɣ	86 (25–250; 9288)
Ribociclib	46 (24–180; 116) ɣ	50 (16–160; 1728)

# Multiple outcomes can be reported for the same report. ɣ If calculated considering only reports in which thromboembolic events were reported alone (i.e., without concomitant events): 163 (94–277; 16), 141 (56–375; 202), 77 (31–326; 43) for abemaciclib, palbociclib and ribociclib, respectively. ± Breast cancer metastatic, metastases to bone, and breast cancer stage iv reported in the therapeutic indications. ‡ Including aromatase inhibitors and/or fulvestrant. € based on the Medical Dictionary for Regulatory Activity high-level group term “anaemias nonhaemolytic and marrow depression.” $ Based on co-reported cardiovascular drugs or cardiovascular indications. Italic: detailed feature.

**Table 2 cancers-13-01758-t002:** Disproportionality analyses.

Outcome	Analysis	CDK4/6 Inhibitors	Abemaciclib	Palbociclib	Ribociclib	Tamoxifen
		*n*, ROR (95% CI)	*n*, ROR (95% CI)	*n*, ROR (95% CI)	*n*, ROR (95% CI)	*n*, ROR (95% CI)
TEs (overall)	Exploratory	1722, 0.90 (0.86–0.94)	112, 0.90 (0.75–1.09)	1359, 0.86 (0.81–0.91)	254, **1.18 (1.04–1.34)**	411, **1.30 (1.18–1.44)**
	Consolidated	1722, 0.89 (0.85–0.93)	112, 1.07 (0.88–1.3)	1359, 0.83 (0.78–0.87)	254, **1.29 (1.14–1.47)**	NA
VTE	Exploratory	659, **1.51 (1.39–1.63)**	51, **1.80 (1.36–2.38)**	512, **1.41 (1.29–1.54)**	96, **1.93 (1.58–2.37)**	198, **2.73 (2.37–3.14)**
	Consolidated	659, **1.09 (1.01–1.18)**	51, **1.54 (1.17–2.04)**	512, 0.99 (0.91–1.09)	96, **1.53 (1.25–1.88)**	NA
ATE	Exploratory	355, 0.52 (0.46–0.57)	19, 0.43 (0.27–0.67)	269, 0.47 (0.42–0.53)	69, 0.89 (0.70–1.13)	81, 0.71 (0.57–0.88)
	Consolidated	355, 0.55 (0.49–0.61)	19, 0.55 (0.35–0.86)	269, 0.49 (0.44–0.56)	69, 1.06 (0.83–1.35)	NA
TEs (other)	Exploratory	815, 0.88 (0.82–0.94)	50, 0.84 (0.63–1.11)	664, 0.87 (0.80–0.94)	104, 0.99 (0.82–1.21)	156, 1.01 (0.86–1.18)
	Consolidated	815, 1.01 (0.94–1.08)	50, 1.13 (0.85–1.5)	664, 0.97 (0.90–1.05)	104, **1.25 (1.02–1.51)**	NA

ATE: arterial thromboembolism; TE: thromboembolic events; VTE: venous thromboembolism. NA: not applicable (see text for details). Bold: statistically significant reporting odds ratio (ROR) (i.e., lower limit of the 95% CI > 1).

**Table 3 cancers-13-01758-t003:** Case-by-case assessment.

TEs (*n*)	Death	Serious	Onset Days Median (25–75%)	Dechallenge/ Rechallenge	Concomitant Drugs (%)							Other Drugs *	Causality Assessment
					Drug-Related Risk Factor				Disease-Related Risk Factors				
					Sex Hormones	GC	ESA	Angiogenesis Inhibitors	Anti- Depressants	Anti- Diabetics	Anti- Thrombotics		
Abemaciclib ATE (19)													
Myocardial infarction (7), TIA (5), stress cardiomyopathy (2)	0 (0%)	16 (84.2%)	205 (96–456)	3 (15.8%)/ NA	0 (0%)	0 (0%)	0 (0%)	1 (5.3%)	1 (5.3%)	0 (0%)	1 (5.3%); VKA	Vitamin D (10.5%)	Highly probable (15.8%)
													Probable (73.7%)
													Possible (10.5%)
Abemaciclib VTE (51)													
PE (30), DVT (17), PT (4)	6 (11.8%)	47 (92.2%)	135 (51–200)	20 (39.2%)/ 1 (2.0%)	0 (0%)	1 (2.0%)	1 (2.0%)	0 (0%)	1 (2.0%)	3 (5.9%)	4 (7.8%); heparins (2)	Denosumab (11.8%)	Highly probable (29.5%)
													Probable (58.8%)
													Possible (11.8%)
Abemaciclib OTE (50)													
Thrombosis (22), cerebrovascular accident (12), cerebral infarction (5)	7 (14%)	37 (74%)	86 (35–242)	7 (14.0%)/ NA	0 (0%)	0 (0%)	0 (0%)	0 (0%)	3 (6%)	3 (6%)	6 (12%); ASA (3), apixaban (2)		Highly probable (10.0%)
													Probable (82.0%)
													Possible (8.0%)
Palbociclib ATE (269)													
myocardial infarction (147), TIA (42), ischemic stroke (11)	39 (14.5%)	248 (92.2%)	224 (72–375)	52 (19.3%)/ NA	0 (0%)	17 (6.3%)	0 (0%)	2 (0.7%)	20 (7.4%)	16 (6.0%)	49 (18.2%); ASA (28), DOACs (15), heparins (8)	Denosumab (10%)	Highly probable (15.6%)
													Probable (68.8%)
													Possible (15.6%)
Palbociclib VTE (512)													
PE (301), DVT (113), PT (87), thrombophlebitis (19), PVT (8)	56 (10.9%)	492 (96.1%)	85 (34–256)	101 (19.7%)/ 2 (0.4%)	5 (1.0%)	37 (7.2%)	3 (0.6%)	14 (2.7%)	42 (8.2%)	29 (5.7%)	109 (21.3%); DOACs (51), ASA (34), heparins (30)	Denosumab (10.2%)	Highly probable (15.2%)
													Probable (65.0%)
													Possible (19.7%)
Palbociclib OTE (664)													
thrombosis (296), cerebrovascular accident (248), cerebral infarction (25)	66 (9.94%)	627 (94.4%)	116 (42–363)	100 (15.1%)/ NA	0 (0%)	41 (6.2%)	0 (0%)	9 (1.4%)	48 (7.2%)	32 (4.8%)	93 (14.0%); DOACs (43), ASA (41), heparins (11)		Highly probable (11.4%)
													Probable (72.4%)
													Possible (16.1%)
Ribociclib ATE (69)													
myocardial infarction (41), TIA (13), AMI (7)	17 (24.6%)	67 (97.1%)	27 (16–134)	21 (30.4%)/ NA	0 (0%)	2 (2.9%)	0 (0%)	1 (1.5%)	3 (4.4%)	2 (2.9%)	8 (11.6%); ASA (6), heparins (3)		Highly probable (29.0%)
													Probable (59.4%)
													Possible (11.6%)
Ribociclib VTE (96)													
PE (56), DVT (21), venous occlusion (6), thrombophlebitis (4)	23 (24.0%)	96 (100%)	105 (37–206)	25 (26.0%)/ NA	0 (0%)	4 (4.2%)	1 (1.0%)	3 (3.12%)	6 (6.3%)	4 (4.2%)	14 (14.6%); ASA/DOACs/ heparins (4)	Denosumab (20.8%)	Highly probable (19.8%)
													Probable (63.5%)
													Possible (16.7%)
Ribociclib OTE (104)													
thrombosis (39), cerebrovascular accident (33), paraplegia (7), cerebral ischemia (6)	12 (11.5%)	102 (98.1%)	45 (18–189)	26 (25.0%)/ 1 (1.0%)	1 (1.0%)	5 (4.8%)	0 (0%)	5 (4.8%)	9 (8.7%)	3 (2.9%)	13 (12.5%), DOACs (5), ASA (4)		Highly probable (21.2%)
													Probable (63.5%)
													Possible (15.4%)

* Most frequently reported drugs different from hormone therapy. AMI: acute myocardial infarction; ASA: acetylsalicylic acid; ATE: arterial thrombo-embolism; DOACs: direct acting oral anticoagulants; DVT: deep vein thrombosis; ESA: erythropoietin-stimulating agents; GC: glucocorticoids; OTE: other thrombo-embolic events; PAT: peripheral artery thrombosis; PE: pulmonary embolism; PT: pulmonary thrombosis; PVT: portal vein thrombosis; TE: thrombo-embolic events; TIA: transient ischemic attack; VTE: venous thrombo-embolism; NA: not applicable.

## Data Availability

The datasets analyzed during the current study are available in the following resource available in the public domain: https://fis.fda.gov/extensions/FPD-QDE-FAERS/FPD-QDE-FAERS.html (accessed on 6 April 2021).

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
