# Peer review of "Thromboembolic Events with Cyclin-Dependent Kinase 4/6 Inhibitors in the FDA Adverse Event Reporting System"

_cancers, 2021, doi:10.3390/cancers13081758_

Round 1

Reviewer 1 Report

The association between cancer and thromboembolic events is well known. Some anticancer treatments are reported to be associated with an excess of thromboembolic events.

However, this risk if most often underestimated and identified lately from usual care, because of lack of systematic assessment in clinical trials assessing the benefit of anticancer drugs.

Authors report data about thromboembolic events submitted to FAERS, a well known international database, during a 15 y period, including an extraction based on thromboembolic terms.

The methodology is rigorous and well described, including a disproportionality analysis and a case by case assessment. Efforts are made to limit the biases related to the risk of thrombosis caused by cancer itself, and to look for the causal role of the drug (CDK4/6 innhibitors).

Comments

In the introduction, authors summarize the available data about thromboembolism and CDK4/6 inhibitors.

In a first approach, VTE was an AESI in clinical trials with abemaciclib, resulting in an accurate assessment of the risk of events in each arm. Such results are not reported for palbiciclib, authors provide results from observational studies.

To make the description of available data for each drug (abemaciclib, palbociclib, ribociclib) clearer, please provide us the data from clinical trials on one side, and from observational studies for each drug.

L88 : Is the study period 2005-2020 or 215-2020 ?

Table 2 : please define the terms : what are ATE ? VTE ? What are thromboembolic events others ?

Figure 1 : Please define the terms : are patients with DVT and PE classified as PE ? Could you provide a figure including additionnally DVT+PE ?

Could you please provide data about characteristics of VTE events (location, proximal or distal ? severity ?)

Authors provide the rate of concomitant antithrombotic drugs please comment on this finding.

Is it possible to have the profile of patients (cnacer, extent, age, cardiovacsular comorbiidities)

Auhtrs could commen on the delay of occurrence of the event as compared with the introduction of the anticancer drug.

In the discussion, authors propose the use of palbo and ribo in paients with CV comorbidities. How can we say that ? to my knowledge, thrombosis was not an AESI in pivotal tirals for these drugs. Additionnally, even if abemaciclib is associate with an increased risk of VTE, I don’t understand the link with patients with CV comorbidities (which ones ?) and those with QT liability.

This sentence is not supported by presented data and the conclusion are not supported by the increased risk of VTE with abemaciclib.

Reviewer 2 Report

Dear editor,

the paper, entitled with “Thromboembolic Events with Cyclin-Dependent Kinase 4/6 Inhibitors in the FDA Adverse Event Reporting System” investigates the reporting rate of adverse thromboembolic events in CDK4/6-inhibitor treated patients.

The study is well conducted and delivers statistical data on this important adverse event. There are some major issues, that limit the overall impact of such an analysis. However, the observed association might be meaningful for the oncologists. The authors should give more information on possible limitations in the abstract in order to simplify the interpretation for the reader.

  • The interpretation ‘possible’ or ‘highly probable’ was done by the authors, based on a definition tool from the WHO. Even the treating physician could only speculate on that. Patients individual history and other important factors are not reported in these databases. The interpretation of such data is therefore limited.
  • The reporting of thrombosis in cancer patients is highly biased due to the fact that it is a high-risk patient cohort per se.
  • As mentioned by the authors in the discussion, the data are not suitable to get insights on incidence rate. This makes it explicit difficult to interpret the data and how meaningful they are for the individual decision. This limitation should be mentioned in the abstract.

Reviewer 3 Report

cancers-1143806 deals with a very interesting topic on the association between thromboembolic events and CDK4/6 inhibitors using a pharmacovigilance database and disproportionality analyses. Authors found an association between these AEs and CDK4/6 inhibitors.

Major comments:

-authors claimed that this study using the FAERS database allows to highlight a causal association. Unfortunately, it is completely impossible and wrong. The main limitation of pharmacovigilance is not being able to confirm a causal association, due to the large number of confounding factors not explored. Contrary to what authors said, the reference 18 highlighted that the correlation between risks obtained from meta-analyses and disproportionality analyses is very weak and therefore it is impossible to conclude that pharmacovigilance database are “potential tool for risk assessment”. The Case-by-Case Assessment does not in any way confirm a causal association.

-the associations found between thromboembolic events and CDK4/6 inhibitors are very weak even though authors used very sensitive and favorable conditions for the emergence of a significant signal (no adjusted analysis, remove of competing AEs potentially masking the identification of thromboembolic events…). The present significant but weak associations found may not remain significant with the use of less sensitive conditions, which reinforces the fact of not being able to affirm a causal relationship between thromboembolic events and CDK4/6 inhibitors in this present study.

-Authors must absolutely consider adjusting their ROR on several confounding factors including at least: other anticancer drugs, other drug associated with thromboembolic events, drugs of interest (anticoagulants…), concomitant AEs, age.

-The dechallenge, as a causality parameter, is probably difficult to interpret in the context of thromboembolic events. For example, in case of pulmonary embolism, an anticoagulant is prescribed and is able to completely treat the pulmonary embolism, even if in case of CDK4/6 inhibitors continuation.

Minor comments:

-FAERS is not a worldwide pharmacovigilance database but a US one. Vigibase is the worldwide pharmacovigilance database.

-Regarding ROR calculation, did authors used concomitant or/and suspected reports?

-Authors have chosen SMQ to search cases. This is debatable as SMQ include some AE, probably not in line with the present study. Probably the main proportion of cases are concentrated in some Preferred Terms such “pulmonary embolism”, that are maybe more clinically relevant.

-A high proportion of reports come from consumers. please discuss this point.

-A high proportion of cases are co-reported with a cancer progression that is a well-known cause of thromboembolic events. Could authors discuss this point please?

-Stage of cancer development is not mentioned (metastasis or localized).

-Concomitant anticancer drugs are not available. CDK4/6 inhibitors are frequently prescribed with hormonotherapies.

-Concomitant AEs of interest, such as anemia (that are associated with thromboembolic events) are not available.

-lines 242-243: Khorana score is not validated for the evaluation of the thromboembolic risk of oral anticancer drugs and it does not consider the metastasis status. Maybe authors could more modulate this statement.
